# Identifying and Ranking the Dimensions of Urban Resilience and Its Effect on Sustainable Urban Development in Tongdejie, China

## Jiarong Xie

Faculty of innovation and Design, City University of Macau, Macau 999078, China; xjiarong888@163.com

**Abstract:** Urban resilience strengthens urban sustainability and leads to sustainable solutions in the process of promoting sustainable development. Paying attention to the benefits of urban resilience to strengthen sustainable urban systems is vital to achieve our desired future. This study aims to identify and classify the key indicators of resilience in Tongdejie, a residential area in Guangzhou, China, as well as to examine and compare these indicators with sustainable development indicators. Fuzzy AHP was used to rank the most important indicators in Tongdejie. The results showed that the first three important indicators were economic indicators, and the economic dimension with a weight of 0.41 was higher than the other four indicators. Then, social and cultural indicators took second place with a weight of 0.194, and the management and institutional indicators took third place with a weight of 0.194. Structural–physical and environmental dimensions were ranked fourth and fifth, respectively. From the obtained results and their comparison with the sustainable indicators, in addition to ranking the importance of these indicators and incorporating the research related to urban construction development indicators, it can be concluded that these two concepts have a direct relationship with each other. In order to attain a desired and resilient urban future, it is important to pay attention to the indications and advantages of resilience. This leads to the development and stability of urban systems.

**Keywords:** identification; ranking; dimensions of resilience; sustainable development; Tongdejie

## 1. Introduction

It is estimated that by 2050, over half of the Earth's population will be displaced into urban areas and vast cities due to the increase in population. To properly manage citizens' needs and provide advanced services, an advanced infrastructure is needed [1]. Cities are complex systems that become very vulnerable when one of their subsystems is damaged or has difficulty adapting to crises [2]. These regions are sensitive to global sustainability, so sustainability needs to be increased in these areas, and they are location where changes in energy efficiency, adaptation to climate change, and social innovation occur [3]. The latest developments indicate the possibility of various crises such as economic crises or climate change in these places, and one of the obvious forms of crises is a serious and significant increase in demand due to an increase in urban population, whereby cities must provide products and services with environmental and social effects to the interior regions of countries [4]. By 2050 AD, 233 million very poor people will live in 43 countries and will be at risk of experiencing crises [5].

The progress of resilience in facing political, socioeconomic, and environmental adversities has attracted the attention of academic and decision-making communities. Resilience, especially for cities, has become an important factor to combat climate change [6].

The relationship between resilience and urban sustainability is interesting from a theoretical and empirical point of view. Urban resilience strengthens urban sustainability, which leads to sustainable solutions in the process of promoting sustainable development. Paying attention to the benefits of urban resilience to strengthen sustainable urban systems

is a vital factor to achieve our desired future [7]; on the other hand, urban sustainability is related to the paradigm of sustainable urban development, the antecedent of which goes back to our common future. In some people's opinions, sustainability and resilience are concepts that can be interchangeably used, and in the opinion of others, resilience is one of the purposes of sustainability. It is even considered a main factor and a factor in its consolidation. Based on Lung, in the last two decades following the economic crises of the world in 3000 AD and the economic, political, and social crises in European countries, the concept of resilience, especially urban resilience as it pertains to the concept, research in urban planning, policy and practice, and planning for resilient regions and cities, was developed by the Association of University Schools of Planning and the European School of Planning Association in 2013 and was recognized by urban science societies in the United States and the European Union [8]. In 2014, the Resilience Forum was held in Montpellier, France. A large number of government managers, researchers, and urban planners have participated in urban resilience studies. Many scientific organizations such as the Resilience Association, Resilience Organization, and Resilient City Organization were formed at the global level [8]. It is necessary to know how the theory of resilience was developed in order to comprehend the concept of urban resilience, even though it was used in psychology and engineering. When it comes to the global literature about environmental changes, researchers typically refer to resilience by mentioning Holling's name. The concept of resilience was defined by Holling as the capability of systems to maintain their basic functions in the face of disruptions [9]. He distinguishes between static (passive) engineering resilience (referring to the ability of a system to return to its previous state) and dynamic ecological resilience (referring to the ability of key functions to be maintained during disturbances) by describing ecosystems as multiple stable states [9].

China is urbanizing at a high rate. This is probably one of the biggest human resettlement experiments in history. In the period from 1978 to 2012, only a part of the country's population lived in cities. Over time, the population of cities has grown by 32.7%; that is, it has increased from 17.9% to 52.6%. If the current trends continue, China's urban population could reach one billion folks in the next two decades; these are uncharted waters, but China has a plan. In March, with the goal of achieving an annual population growth of 1 percent, the government published a new type of national urban development plan to increase the annual population growth to 60 percent by 2020 [10]. This comprehensive and ambitious scheme encompasses almost every imaginable aspect of urbanization, including rural-to-urban migration and integration with spatial distribution, links among the cities, sustainable development, institutional arrangements, and execution. It sets numerical targets ("Government target"), and it also serves as a guiding principle on sustainability and uses a people-centered approach, whereby more attention is paid to welfare and well-being, which creates a significant and positive change from the current economic focus on land development. Additionally, another purpose is to correct the present problems associated with rapid urbanization that have been occurring in the last three decades. Although the right national strategy is necessary, it cannot be sufficient. It is the local practices that play an effective role in the success or failure of China's urban planning [8].

The concepts of sustainability and resilience are two interwoven concepts, and both are known as system capabilities (urban, social, ecological) that can provide desirable developments [11]. The variety of connections between these two concepts was examined in the literature, and sustainability was presented as a normative concept to promote justice, while resilience was studied as a descriptive, favorable, and sometimes even unfavorable concept [12]. However, combining the aspects of sustainability and resilience provides strategies that make these two concepts work with full efficiency: a city cannot be resilient without being sustainable and vice versa [13].

Apart from the advantages of separately applying resilience in urban contexts, the simultaneous application of these two concepts in urban development will provide multiple benefits for urban socioeconomic systems. There are several reasons for this when it comes to the principles and procedures of planning and urban management; additionally,

the concepts should be assessed by primarily using tools and guides to link these two meanings [14]. This solidarity in the urban environment and the field of sustainable development is a strength for investing in the development, and the sustainability of urban systems will provide benefits [15]. In the process of development, nonresilient but sustainable urban systems lose the benefits they have accumulated, and owing to the lack of resilience against risks, they may even return to their predevelopment state. Our research uses this meaning and examines resilience in China's Tongdejie region according to five economic, social, and cultural dimensions; environmental factors; physical structure; and management issues.

*Literature Review*

The concept of resilience has been integrated into sustainability via several quantitative methods. By using a probabilistic risk analysis, Walker et al. (2010) [16] incorporated resilience into sustainability quantification by defining sustainability as the ability to achieve a nondecreasing level of welfare. A metric-based framework was proposed by Jarzebski et al. (2016) [17] to measure economic, environmental (natural capital), and social (sociocultural capital) sustainability. Resilience indicators include trust in the local government, traditional farming practices, agroforestry practices, and forest cover percentages. Based on Milman and Short (2008) [18], water system sustainability can be built using indicators such as water supply over the next 50 years, quality of service (e.g., chlorinated pipes or public wells), and financial risk to water providers.

Some studies include other framework components in addition to resilience as a component of sustainability. Based on Saunders and Becker (2015) [13], risk management contributes to sustainability through resilience. Research on resilience includes risk in various ways [19–21]. Using a risk management framework to conduct case studies of earthquake-prone communities in New Zealand, Saunders and Becker (2015) [13] concluded that lowering risk leads to greater resilience and sustainability. Similarly, Seager (2008) [22] presented resilience as one perspective among the four aspects of sustainability, which include security, reliability, and renewal. Throughout this spectrum, sustainability moves from a state of security or stability (a state in which the status quo remains unchanged) to one in which rapid change and an all-encompassing reorganization are enabled (Seager, 2008) [22].

Zeng et al. [23] identified several indicators in the main dimensions (social, economic, and environmental) of urban sustainability. A systematic literature review was conducted with PRISMA using the literature from 1 January 2001 to 30 November 2021. The results showed that sustainability and resilience are related paradigms that emphasize the capacity of a system to move toward desirable development paths. Resilience and sustainability are fundamentally related to maintaining social health and well-being within a broader framework of environmental change. A study conducted by Cores et al. [24] examined the resilience strategies developed by 100RC cities and found that they are significantly aligning their efforts with the SDGs as part of global development policies. To illustrate how resilience strategies are developed using the tools and methods of 100RC, such as the City Resilience Framework (CRF) and the City Resilience Index (CRI), and how they align with the sustainability goals, the city of Cape Town was used, which represents the lessons learned from the post-2015 Cape Town network global-policy nexus. An evaluation index system for urban resilience was developed by Huang et al. [25] in 2022, and it covered the economic, environmental, social, and infrastructure aspects of Tongdejie, China. From 2005 to 2018, 138 cities were assessed based on the evaluation index using the entropy weight method. Despite the eight urban masses' generally low resilience levels, an upward trend in resilience was noted. Based on panel data from 2012 to 2017, Shi et al.'s [26] study of 282 Chinese cities employed the entropy approach to compute the urban resilience index. Their study explored the spatial characteristics of urban resilience using a spatial hot–cold spot model, constructed the characteristics of the spatial network of urban resilience using the gravity model, and analyzed the spatial network of urban resilience using social

network analysis. The findings revealed that Chinese cities' urban resilience had gradually improved, and there was a geographical accumulation effect with significant changes in hot spots but insignificant changes in cold spots. Suarez et al. [27] studied 50 Spanish cities to find a methodological framework to measure urban resilience. The results showed that most of the centers of Spanish provinces had low resilience. They proposed resilience indices to measure resilience. Yang et al. [15] examined the Chengdu-Chongqing economic circle resilience and effect. The TOPSIS entropy method was used to evaluate the level of urban resilience of the Chengdu-Chongqing economic circle, and the fuzzy set qualitative comparative analysis approach was used to analyze the configuration of contributing factors. The research showed that the overall level of urban resilience was relatively low, with more than 70% of areas below 0.3.

In 2018, Masnavi et al. [28] introduced urban resilience indicators to the relationship between urban form and urban resilience via a specific approach. More studies should be conducted on spatial morphology and urban spatial structures according to a review of the literature on urban resilience. The bulk of research subjects focused on environmental aspects and natural hazard mitigation, such as global warming and climate change. Moreover, it is necessary to conduct further research on the criteria used to measure urban resilience, particularly due to spatial–spatial aspects. In a 2019 article, Ribeiro and Gonçalves [3] reviewed the scientific and technical literature on urban resilience and highlighted its definitions, dimensions, applications, contexts, features, challenges, and opportunities. As a result of these issues, the purpose of this research was to develop a systematic approach and a clear view of urban resilience for the purpose of strengthening and building urban communities against new disturbances. According to their research, urban resilience is the result of four basic components: resistance, recovery, adaptation, and transformation. The five dimensions of urban resilience are natural, economic, social, physical, and institutional.

In a review study in 2019, Cariolet et al. [29] studied proposed methods and approaches to mapping urban resilience against disasters, and one of the most important results of this research showed that the selection of variables and indicators to measure and map resilience is often a function of data availability and reliability. In 2021, Jamali et al. [30] evaluated the resilience of Tehran city against risks by using the DNAP modeling method based on GAS. The four influential dimensions of resilience were selected using the Delphi method. This study utilized three dimensions: the environmental dimension, the physical dimension, and the socioeconomic dimension, each of which included subcriteria. As a result of this study, disasters and natural hazards in the environmental dimension, urban infrastructure in the physical dimension, and employment rate in the socio-economic dimension were proven to be the most important factors that affect urban resilience.

Both theoretically and empirically, there is an interesting relationship between resilience and urban sustainability. The resilience of an urban area contributes to its sustainability, which leads to sustainable solutions in the process of promoting sustainable development. To achieve our desired future, it is necessary to pay attention to the benefits of urban resilience to strengthen sustainable urban systems [31]; additionally, urban sustainability is closely tied to the paradigm of sustainable urban development that was introduced in our joint report. Sustainability and resilience are viewed by some as interchangeable concepts, while resilience is viewed by others as one of the purposes of sustainability and as its underlying condition.

The preparation and training of citizens on how to correctly behave in the face of crisis, attention to the degree of vulnerability of the city, the provision of infrastructure and manpower, and greater resilience and flexibility in the face of risks will lead to stability in urban spaces since cities are places of population density and are man-made phenomena. In the absence of necessary conditions of resilience and stability, a lot of financial and human losses will be imposed on the urban body. Our research focuses on a residential area in Guangzhou that contains a wide range of housing types, from low-income housing to urban villages to old city neighborhoods [32].

The most important reason for choosing this region was to study and identify the indicators used to select a specific region with different conditions in big cities, which are the subjects of the majority of the research conducted in this field, and to examine the difference between the indicators and their importance. Moreover, the second innovation of this plan was to compare the indicators with the criteria and purposes of sustainably developing cities, and this comparison dealt with the understanding and importance of indicators of sustainability and resilience together. Finally, the ranking of these indicators indicates the development direction, which policymakers and city officials can use, and more attention will need to be given to eliminate obstacles to urban resilience and sustainability.

The outline of this study is as follows: First, we identify the development criteria within five dimensions based on the literature and the opinions of elites, and then we rank them. Then, we analyze the relationship between each of the dimensions and sustainable development indicators as a final objective. We then compare our criteria with those previously published by other researchers in the field of sustainability, and we analyze the importance of these indicators from a sustainability perspective. These results can provide policymakers and city officials with an indication of the city's development direction and a greater awareness of the obstacles to urban resilience and sustainability.

## 2. Methodology

Regarding the classification of scientific research, this research is practical research. This research aims to identify and rank the dimensions of urban resilience in Tongdejie based on sustainable development indicators. Regarding the data collection method, the current research is descriptive, the purpose of which was to know more about the existing development conditions and also to help the decision-making process. The library method (domestic and foreign books and articles) was used, and in order to analyze the data, a combination of decision-making methods was used in conditions of uncertainty, and finally, the indicators were weighted. The fuzzy hierarchical analysis process (fuzzy AHP) was used due to the hierarchical nature of the research factors. The statistical population of this research was experts in the construction industry, including civil engineers, architects, and urban planners, who were familiar with the nature of the research topic. We then used the snowball sampling method, and 12 experts in this field were recruited. The researchers conducting this study primarily interviewed city planning and municipal employees as well as construction engineers. The process began with a list of experts. Then, 12 experts were purposefully selected based on their education and work experience in order to determine how their degree of work experience related to their chosen criteria. In general, the steps related to the research are given in Figure 1.

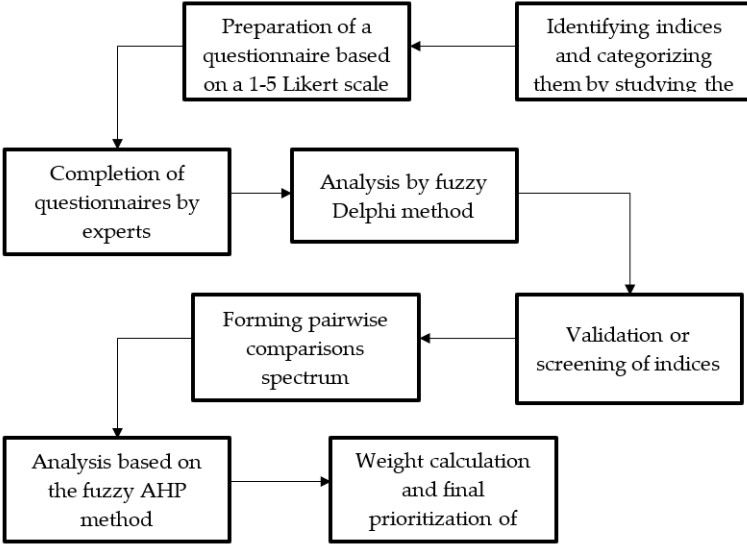

**Figure 1.** General stages of research.

### 2.1. Fuzzy Theory

People's judgments about preferences are often not transparent enough to be represented by an exact numerical value; additionally, fuzzy logic is useful to solve problems that have ambiguity and uncertainty. Fuzzy theory was first proposed by Lotfizadeh (1965) to reconcile the uncertainty surrounding human understandings of the model [33]. Fuzzy numbers are represented by the symbol "(·)" above the number. A triangular fuzzy number is shown in Figure 2 [34].

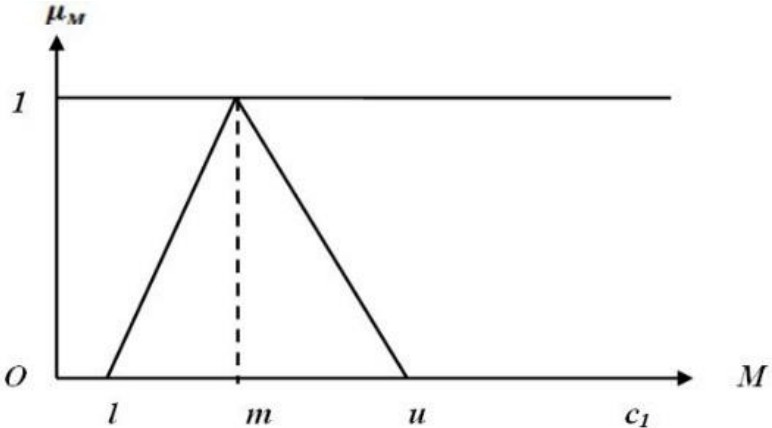

**Figure 2.** Representation of triangular fuzzy number.

Triangular fuzzy numbers are presented as ($l$, $m$, $u$). The parameters $m$, $l$, and $u$ are, respectively, the smallest possible expected value, the most likely expected value, and the largest possible expected value.

### 2.2. Fuzzy Delphi Method

Dalkey and Helmer first introduced the Delphi method in 1963, which is a survey method based on expert opinions. Nameless answers, regulated repetition and feedback, and statistical group reactions are this method's three primary features. This method is a methodical strategy that is used to compile and organize the knowledgeable opinions of a group of specialists about a certain subject or problem [35]. In general, the judgment of experts cannot be expressed and interpreted as definitive quantitative numbers in most real situations. In other words, when modeling real-world systems, it does not seem appropriate to rely on data and definite numbers due to the uncertainty and ambiguity of the judgments of decision makers. To overcome this problem, the "fuzzy sets theory" is used as a suitable tool to deal with ambiguity and uncertainty of the decision-making method, which was presented by Lotfizadeh in 1965 [36]. Hence, the Delphi method was used. Fuzzy logic was used to check and screen the identified indicators. This method was presented by Ishikawa et al., and it is a combination of Delphi method and fuzzy set theory [37].

The steps of performing the fuzzy Delphi method are:

1.  Identifying the research indicators after comprehensively reviewing the theoretical foundations of the research based on the target area.
2.  Collecting the opinions of decision-making experts: in this stage, after identifying the criteria, a decision-making group consisting of experts in fields related to the research topic is formed, and questionnaires are used to determine the relationship between the identified indicators and main ones. The topic of research and screening is presented to them by using the linguistic variables of Table 1 to express the importance of each indicator [38]. Triangular fuzzy numbers are used.

**Table 1.** Linguistic expressions and fuzzy Delphi numbers.

| Linguistic Expressions | Triangular Fuzzy Numbers |
|---|---|
| Very low | (0, 0, 0.25) |
| Low | (0, 0.25, 0.5) |
| Medium | (0.25, 0.5, 0.75) |
| High | (0.5, 0.75, 1) |
| Very high | (0.75, 1, 1) |

Verification and screening of indicators: This study was conducted by comparing the value of the acquired value of each indicator with the threshold value of *S*. The threshold value was determined by the decision maker's mental inference, and the number of factors screened are directly affected by these factors. The threshold value cannot be easily and legally determined. The value of 0.7 is considered the threshold value [38]. For this purpose, first the triangular fuzzy values of experts' opinions should be calculated, and then their fuzzy average should be calculated to calculate the average of n respondents' opinions. The fuzzy number τ is calculated for each of the indicators using Equations (1) to (4) [39].

$$\widetilde{\tau}_{ij} = \left(a_{ij}, b_{ij}, c_{ij}\right), \; i = 1, 2, \ldots, n \; j = 1, 2, \ldots, m \tag{1}$$

$$a_j = \sum_{j=1}^{n} \frac{a_{ij}}{n} \tag{2}$$

$$b_j = \sum_{j=1}^{n} \frac{b_{ij}}{n} \tag{3}$$

$$c_j = \sum_{j=1}^{n} \frac{c_{ij}}{n} \tag{4}$$

$$defuzzy = \frac{a + b + c}{3} \tag{5}$$

In the above equations, the index *i* refers to the expert and the index *j* refers to the decision-making index. Furthermore, the diphase value of the average fuzzy number is obtained from Equation (5) [39].

*2.3. Buckley's Geometric Mean Method*

In this method, Buckley's geometric mean technique is used to calculate the relative weights in fuzzy pairwise comparisons (Mathew et al., 2020) [40]. The steps of this method are given below. It is assumed that $\widetilde{P}_{ij}$ is a set of decision makers' preferences about one index compared to the other indices.

The matrix of paired comparisons is formed as follows:

$$\widetilde{A} = \begin{bmatrix} 1 & \widetilde{P}_{12} & \widetilde{P}_{1n} \\ \widetilde{P}_{21} & 1 & \widetilde{P}_{2n} \\ \widetilde{P}_{n1} & \widetilde{P}_{n2} & 1 \end{bmatrix} \tag{6}$$

where *n* is the number of related elements in each row. The fuzzy weights of each index of the pairwise comparison matrix are obtained by using Buckley's geometric mean method. The geometric mean value of the fuzzy comparisons of index *i* to each index is obtained from Equation (1):

$$\widetilde{r}_i = \left(\prod_{j=1}^{n} \widetilde{P}_{ij}\right)^{1/n} i = 1, 2, 3, \ldots, n \tag{7}$$

Then, the fuzzy weight of the *i*-th index is shown by a triangular fuzzy number, which is calculated by using Equation (2):

$$w_i = r_i \otimes (r_1 \oplus r_2 \oplus \ldots \oplus r_m)^{-1} \tag{8}$$

After calculating the fuzzy weight factors, the weights are diphased using Equation (3) and are then normalized. For normalization, each nonphase weight is divided by the sum of the nonphase weights:

$$w_{crisp} = \frac{l + 2m + u}{4} \tag{9}$$

The verbal expressions and triangular fuzzy numbers listed in Table 2 were used to calculate the weight in the pairwise comparisons.

**Table 2.** Verbal expressions and fuzzy numbers for pairwise comparisons.

| Code | Priorities | Fuzzy Equivalent of Priorities | | |
|------|-----------|--------------|--------------|--------------|
| | | Lower Limit (L) | Middle Limit (m) | Upper Limit (u) |
| 1 | Equal importance | 1 | 1 | 1 |
| 2 | Equal importance to relatively more important | 1 | 2 | 3 |
| 3 | Relatively more important | 2 | 3 | 4 |
| 4 | Relatively more to very important | 3 | 4 | 5 |
| 5 | Great importance | 4 | 5 | 6 |
| 6 | High importance to very important | 5 | 6 | 7 |
| 7 | Very important | 6 | 7 | 8 |
| 8 | Very much to absolutely more important | 7 | 8 | 9 |
| 9 | Absolutely more important | 8 | 9 | 10 |

## 3. Results

First, in light of a literature review and ELIT's opinion, the dimensions of urban resilience appear to be as follows: social, economic, and environmental. Then, after using the fuzzy Delphi method and confirming and screening 12 experts, a questionnaire was distributed to the experts who rated each criterion on a scale from 1 to 5. The indicators were made and the results are given in Table 3. The results showed the rejection of five indicators that obtained a nonfuzzy score of less than 0.7, and 23 indicators were approved. The corresponding indicators and components used in this research were created based on the opinions of elites and previous research, namely the research of Norris et al. [41], Patel and Nosal (2016) [42], Cimellaro (2016) [43], Romero-Lankao et al. (2016) [44], Zhang and Li (2018) [7], Ajibid (2017) [45], and Borsekova [46].

**Table 3.** Fuzzy score, subcriterion, nonfuzzy score, and status.

| Criterion | Subcriterion | Fuzzy Score | Nonfuzzy Score | Status |
|-----------|-------------|-------------|----------------|--------|
| Economical | Economic participation of women | (0.5, 0.75, 0.938) | 0.729 | Confirm |
| | Poverty | (0.563, 0.813, 0.958) | 0.778 | Confirm |
| | Home ownership | (0.521, 0.771, 0.958) | 0.75 | Confirm |
| | Employment status and income | (0.542, 0.792, 0.979) | 0.771 | Confirm |
| | Economic dynamism and diversity | (0.5, 0.729, 0.875) | 0.701 | Confirm |
| | Insurance coverage | (0.563, 0.813, 1) | 0.792 | Confirm |
| | Dependence of employment on a specific sector | (0.354, 0.563, 0.771) | 0.563 | Reject |

**Table 3.** *Cont.*

| Criterion | Subcriterion | Fuzzy Score | Nonfuzzy Score | Status |
|---|---|---|---|---|
| Social and cultural | Justice and social equality | (0.5, 0.75, 0.958) | 0.736 | Confirm |
| | Literacy and awareness | (0.521, 0.771, 0.938) | 0.743 | Confirm |
| | Social vulnerability | (0.521, 0.771, 0.958) | 0.75 | Confirm |
| | Social stability | (0.229, 0.417, 0.646) | 0.431 | Reject |
| | The level of people's participation | (0.521,0.771,0.938) | 0.743 | Confirm |
| | Access to transportation and health services | (0.521, 0.771, 0.958) | 0.75 | Confirm |
| Environmental | Environmental hazards | (0.479, 0.729, 0.938) | 0.715 | Confirm |
| | Energy consumption (water, electricity, gas, etc.) | (0.479, 0.729, 0.938) | 0.715 | Confirm |
| | Quality and construction materials | (0.521, 0.771, 0.938) | 0.743 | Confirm |
| | Pollution | (0.5, 0.75, 0.938) | 0.729 | Confirm |
| | Environmental diversity | (0.229, 0.458, 0.708) | 0.465 | Reject |
| | Environmental sustainability | (0.583, 0.833, 1) | 0.806 | Confirm |
| Physical–structural | Land uses | (0.521, 0.771, 0.979) | 0.757 | Confirm |
| | The texture and body of the city | (0.521, 0.771, 0.958) | 0.75 | Confirm |
| | City form | (0.563, 0.813, 0.938) | 0.771 | Confirm |
| | Buildings and historical buildings | (0.375, 0.625, 0.875) | 0.625 | Reject |
| | Neighborhood cohesion | (0.542, 0.792, 0.958) | 0.764 | Confirm |
| Institutional and managerial | Integrated management between urban organizations | (0.542, 0.792, 0.979) | 0.771 | Confirm |
| | Public safety | (0.521, 0.771, 0.938) | 0.743 | Confirm |
| | Urban development planning and technology | (0.563, 0.813, 0.958) | 0.778 | Confirm |
| | Organizational flexibility | (0.333, 0.583, 0.833) | 0.583 | Reject |

### 3.1. The Results of the Fuzzy AHP Method

A pairwise comparison of the main criteria and then the subcriteria was conducted in this section based on the research indicators that were confirmed in Table 3. After answering the paired comparisons, the inconsistency rates of the tables were calculated, all of which were smaller than 0.1, which indicates that the stability and reliability of the paired comparisons was acceptable. Then, the responses were integrated using the geometric mean method in the form of comparisons. The merged pair is given below. The weights of the pairwise comparisons were calculated using the Buckley's geometric mean method.

3.1.1. Formation of Paired Comparisons

In this section, the main criteria are given as examples of pairwise comparisons. These pairwise comparisons were made by experts based on the spectrum from phases 1 to 9 given in Table 1, and then they were integrated by using the geometric mean method, which is given in Table 4.

**Table 4.** Pairwise comparisons of criteria (inconsistency rate: 0.05).

|  | Economical | Social and Cultural | Environmental | Physical–Structural | Institutional and Managerial |
|---|---|---|---|---|---|
| Economical | (1, 1, 1) | (1.587, 2.556, 3.446) | (1.414, 2.196, 3.086) | (1.799, 2.589, 3.592) | (3.456, 4.637, 5.731) |
| Social and cultural | (0.29, 0.391, 0.63) | (1, 1, 1) | (1.353, 1.73, 2.184) | (1.389, 1.851, 2.326) | (1.017, 1.208, 1.407) |
| Environmental | (0.324, 0.455, 0.707) | (0.458, 0.578, 0.739) | (1, 1, 1) | (0.558, 0.704, 0.888) | (0.491, 0.629, 0.795) |
| Physical–structural | (0.278, 0.386, 0.556) | (0.43, 0.54, 0.72) | (1.126, 1.42, 1.793) | (1, 1, 1) | (1.178, 1.514, 1.854) |
| Institutional and managerial | (0.175, 0.216, 0.289) | (0.711, 0.828, 0.984) | (1.258, 1.59, 2.035) | (0.539, 0.661, 0.849) | (1, 1, 1) |

3.1.2. Calculation of Fuzzy and Normal Weights

This step involves calculating the geometric mean of each fuzzy number in each row of the table based on Equations (1) and (2) and then multiplying the geometric mean by the sum of the geometric means to obtain a fuzzy weight. Fuzzy weights are calculated by using the equation (l + 2 m + u)/4, and the nonfuzzy weights are normalized by dividing them by the sum. Moreover, Table 4 presents the economic criteria in the following way:

The geometric mean of the rows of Table 4 is calculated as follows:

*The first line of the geometric mean*
$$= [(1, 1, 1) \times (1.587, 2.556, 3.446) \times \ldots \times (3.456, 4.637, 5.731)]^{\frac{1}{5}}$$
$$= (1.694, 2.321, 2.938)$$

In a similar way, these calculations are performed for the other lines, and the results are given in the second column of Table 5 for all the lines. This gives us the sum of all the geometric means, which is equal to 4.412, 5.626, and 7.034, and then the fuzzy weight of each criterion is equal to the geometric mean of the line of that criterion divided by the sum of the geometric means. For example, for criterion C1, the fuzzy weight is as follows:

fuzzy weight of economic criterion = ((1.694,2.321,2.938))/((4.412,5.626,7.034)) = (0.241,0.413,0.666)

**Table 5.** Fuzzy and nonfuzzy weight of the main criteria.

| Criterion | $\left(\left(\prod_{j=1}^{n}\widetilde{P}_{ij}\right)^{1/n}\right)$ Geometric Mean | Fuzzy Weight (W)~ | Nonfuzzy Weight | Normal Weight |
|---|---|---|---|---|
| Economical | (1.694, 2.321, 2.938) | (0.241, 0.413, 0.666) | 0.433 | 0.41 |
| Social and cultural | (0.889, 1.086, 1.351) | (0.126, 0.193, 0.306) | 0.205 | 0.194 |
| Environmental | (0.527, 0.651, 0.819) | (0.075, 0.116, 0.186) | 0.123 | 0.117 |
| Physical–structural | (0.692, 0.852, 1.059) | (0.098, 0.151, 0.24) | 0.16 | 0.152 |
| Institutional and managerial | (0.61, 0.716, 0.868) | (0.087, 0.127, 0.197) | 0.134 | 0.127 |
| $\sum\left(\prod_{j=1}^{n}\widetilde{P}_{ij}\right)^{1/n}$ | (4.412, 5.626, 7.034) | | | |

The same operation is performed for all criteria, and the fuzzy weights are given in the third column of Table 5. Then, each fuzzy weight is defuzzied as follows:

Fuzzy economic weight = (0.241, 0.413, 0.666) => Nonweight fuzzy economic = (0.241 + 2 × 0.413 + 0.666)/4 =0.433. This process is performed for all the criteria, and the results are given in the fourth column of Table 5, Afterwards, each nonphase weight is normalized as follows:

*Non − weighted fuzzy economy* = 0.433 => *normal weighted economy*
$$= 0.433/(0.433 + 0.205 + \cdots + 0.134) = 0.410 \ 0.410$$

As shown in Table 5, first place goes to the economic criterion with a weight of 0.41, second place goes to the social and cultural criterion with a weight of 0.194, and third place goes to the physical–structural criterion with a weight of 0.152. They ranked fourth and

fifth, respectively, for the institutional and managerial criteria with a weight of 0.127 and the environmental criteria with a weight of 0.117.

The final weight of the factors

To calculate the relative weights, pairwise comparisons were performed for the subcriteria. Finally, the final weight of the subcriteria was obtained by multiplying the relative weight of each subcriteria by the weight of the main criterion, which is given in Table 6. Based on this, economic dynamism and diversity won the first rank. Poverty ranked second and employment status and income ranked third. The weights and final priorities of the criteria are shown in Figure 3.

**Table 6.** Relative and final weight of factors.

| Criterion | Criterion Weight | Subcriterion | The Relative Weight of the Subcriterion | The Final Weight of the Subcriterion | The Final Rank of the Subcriterion |
|---|---|---|---|---|---|
| Economical | 0.41 | Economic participation of women | 0.069 | 0.0283 | 15 |
| | | Poverty | 0.265 | 0.1087 | 2 |
| | | Home ownership | 0.074 | 0.0303 | 14 |
| | | Employment status and income | 0.246 | 0.1009 | 3 |
| | | Economic dynamism and diversity | 0.28 | 0.1148 | 1 |
| | | Insurance coverage | 0.067 | 0.0275 | 18 |
| Social and cultural | 0.194 | Justice and social equality | 0.309 | 0.0599 | 4 |
| | | Literacy and awareness | 0.243 | 0.0471 | 7 |
| | | Social vulnerability | 0.214 | 0.0415 | 10 |
| | | The level of people's participation | 0.092 | 0.0178 | 21 |
| | | Access to transportation and health services | 0.142 | 0.0275 | 17 |
| Environmental | 0.117 | Environmental hazards | 0.307 | 0.0359 | 11 |
| | | Energy consumption (water, electricity, gas, etc.) | 0.158 | 0.0185 | 20 |
| | | Quality and construction materials | 0.102 | 0.0119 | 23 |
| | | Pollution | 0.142 | 0.0166 | 22 |
| | | Environmental sustainability | 0.291 | 0.034 | 12 |
| Physical–structural | 0.152 | Land uses | 0.29 | 0.0441 | 9 |
| | | The texture and body of the city | 0.176 | 0.0268 | 19 |
| | | City form | 0.219 | 0.0333 | 13 |
| | | Neighborhood cohesion | 0.315 | 0.0479 | 6 |
| Institutional and managerial | 0.127 | Integrated management between urban organizations | 0.348 | 0.0442 | 8 |
| | | Public safety | 0.22 | 0.0279 | 16 |
| | | Urban development planning and technology | 0.431 | 0.0547 | 5 |

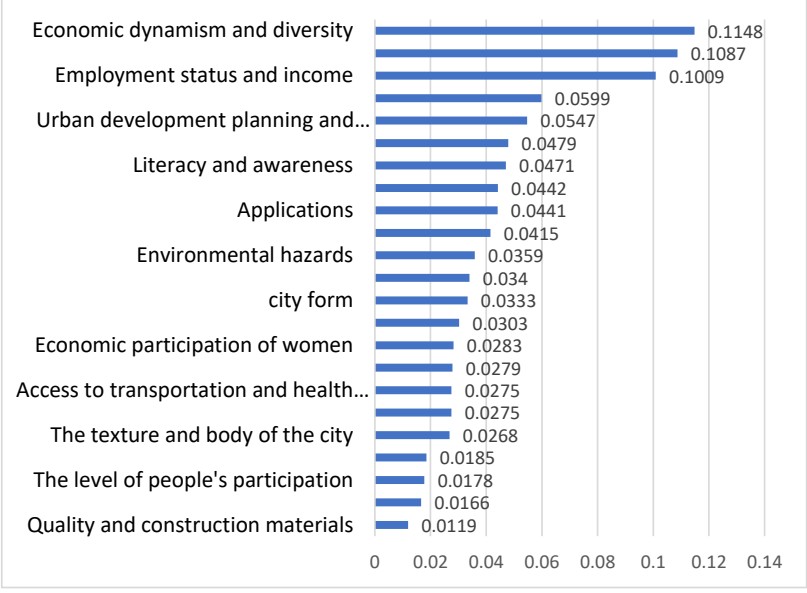

**Figure 3.** Weight and final priority of criteria.

## 4. Discussion

The goals of this study were to first identify the most important indicators of urban resilience and then weight and rank their importance. Important indicators were identified to achieve urban resilience by using the literature and expert's opinions; then, using the fuzzy Delphi method, some of these indicators were removed or integrated. Finally, these indicators were ranked by using a nine-item Likert method, and we analyzed them by using the fuzzy AHP method. The indicators identified in this research were divided into five general categories: economic, social–cultural, environmental, physical–structural, and institutional and managerial.

The economic category included women's economic participation, home ownership, poverty, employment status, income, economic dynamics and diversity, and insurance coverage. Economic resilience refers to the ability to recover from the effects and adjustments of external economic pressures and includes the capacity of the economy to absorb (reduce risk) and recover from incompatible pressures; it also includes the effects of the economy on employment and how frequently employees quit their jobs since temporary losses can turn into structural problems. Therefore, economic resilience is considered the foundation of long-term economic growth capacities and refers to a measure of the economic diversity of society, such as the total employment, the number of commercial establishments, and their ability to function in the wake of crises [47]. The importance of index economic problems when attempting to achieve urban sustainability is not hidden from anyone. So, the emphasis of the World Bank is largely on the economic aspects of sustainable development. In this study, the economic category was ranked first with a weight of 0.41, and the economic obstacles were also the first three most effective obstacles. The economic barrier is one of the most important obstacles of sustainable urban development. Kamranfar et al. [1] in Iran and Ping et al. [48] in Ghana studied the obstacles to the development of sustainable construction in Iran, and economic obstacles were found to be the most influential obstacles. Moreover, Yang and Yang [49] conducted research in Australia and identified economic obstacles as the most important obstacles to overcome. They introduced sustainable construction, and to achieve sustainable and resilient development, their results showed the importance of economic indices.

The social and cultural category includes social justice, equality, literacy, social awareness, social vulnerability, participation rate, and access to transportation and health services. This type of resilience shows the human capacity to zone and plan the future [50]. Social resilience refers to the demographic characteristics of society, including gender, age, ethnicity, disability, socioeconomic status, and other key components such as social capital [51]. Although social capital cannot be measured precisely, it is a sense of community that allows urban groups to adapt to the effects of crises, as well as a sense of place attachment [52]. Our study's findings demonstrated that one of the most useful and significant indicators for establishing urban resilience in China are cultural and social indicators. The relative weight for this category, which was rated second, was 0.192. Furthermore, subindexes of social justice and equality, social literacy, and social vulnerability were ranked 4th, 7th, and 10th, respectively. The social and cultural indices are constant indexes, and they are considered the permanent dimension of urban sustainability. In their research, Kamranfar et al. [1] found a lack of social awareness as the second most important obstacle to achieving sustainable urban development. Furthermore, by conducting research in China in 2019, Wu et al. [53] found that the third important obstacle to achieving green buildings is the lack of environmental awareness.

The environmental category includes environmental risks, energy consumption (water, electricity, gas, etc.), quality and construction materials, pollution, and environmental sustainability. The concept of environmental resilience pertains to the reduction in risks associated with hazards, the return of ecological and environmental services that sustain life after crises, and the use of learning processes to reduce vulnerabilities and future risk. To achieve sustainable construction, it is essential to gain a thorough understanding of the early energy performance of buildings [54]. Alternatively, the use of renewable energy

involves the use of all renewable energy sources, such as the sun, geothermal energy, wind, tides, waves, etc. An example of how biomass is obtained from living or recently living organisms can be considered one of the processes that considers the development of urban sustainability [55], as well as the resilience of cities from an environmental standpoint. In our research, the general category was placed in last place, and it was a less important factor compared to the other four categories.

The physical–structural category includes the uses, texture, and body of the city, city form, buildings, and historical buildings, as well as neighborhood cohesion [56]. The resilience of a place does not only refer to possible operations such as quick responses to critical situations or events such as earthquakes, floods, and other disasters in vulnerable places, but it also includes adaptive strategies and the long-term reduction in crisis effects when facing environmental and social challenges [57]. Buildings are among the preconstruction restrictions in research related to sustainable development indicators, such as in Karji's research in 2020 [58], which identified the main barriers to sustainable construction in the United States. According to our research, this category of indicators ranked third among the five categories that were identified. This category includes uses and forms of the city.

The institutional and management category includes integrated management among urban organizations, public security, and urban development planning and technology. This type of resilience contains features related to risk reduction, planning, and experience of previous disasters. In this context, resilience is determined by the ability to rebuild communities in a way that reduces risks, by obtaining employment for the residents as a risk-reduction effort, by creating organizational links, and by improving and protecting social systems as they exist [59]. In our research, the management institutional indicators ranked fourth with a relative weight of 0.12. The managerial and institutional factor was considered to be the main obstacle in achieving sustainable development. In Karji's research [58] in 2020 in the United States, this category was found to be the second major obstacle in achieving the development of sustainable construction in the United States. Moreover, in Wu's research in China [53], they looked for the obstacles to building development. Green construction and the lack of proper industry policies and guidance in China were the most important obstacles to overcome to achieve sustainable urban development.

Institutional resilience refers to the resilience of the governmental and nongovernmental systems that guide the economy [60]. Resilience is mainly affected by urban governance and infrastructure and the 3084 service levels (resiliency concepts are in two forms) that the government provides [61]. However, Major defines (a) hard resilience, which is the ability to manage structures and organizations that are under pressure, such as by increasing the resilience of a structure with specific strengthening measures that reduce the possibility of their collapse; and (b) soft resilience, which is the ability of systems to absorb and rebuild from the effects of destructive events without making fundamental changes in the function and structure, which refers to the flexibility and adaptability of the system as a whole [62]. Institutional resilience is the ability to react or adapt the social system (organization or society) to the sudden challenges (internal or external or avoid the destructive effects of crises [63].

From the obtained results, it can be seen that the importance of economic indicators in the studied area is very high; economic indicators are one of the main types of indicators both in big cities and in small cities, but they have a higher degree of importance. As previously discussed, the economic index is considered one of the most important indicators in the development of sustainable urban construction. In the previously mentioned article, the authors concluded that resilience is a way to strengthen urban places against environmental crises and hazards, and thus it can be considered one of the pillars of urban sustainability and vice versa. This is true even though many indicators of resilience, such as economic and social dimensions, are common and are a necessary condition for creating a resilient and sustainable city. Otherwise, there is no guarantee that a sustainable city will be resistant to natural and human chaos. Furthermore, resilient cities must be stable against

environmental and human issues. In other words, sustainability is a necessarily continuous concept whose effectiveness is desirable in the long term. Resilience does not require long-term assessment, and though the study area may be a place for ongoing resilience, this is not necessarily the case.

## 5. Conclusions

The vulnerability of cities to natural and social risks and crises is inevitable, and rapid urbanization in the third millennium will cause urban places to have a lack of resources; consume more energy; have higher levels of environmental pollution; experience urban riots; and experience economic, social, and environmental instability. The individuals involved in enacting cities' crisis management plans will surely respond to these issues in the meantime. This sort of crisis is impossible to forecast, and hence there will be many repercussions and consequences because of this. To transition from these disturbance environments and prevent the reduction in urban capacities, when taking a rational approach, resilience and urban sustainability should be combined. According to the results of some studies, despite the theoretical differences between these two concepts of development, in practice, a connection can be established between them. What was confirmed in this research is that there is a positive and meaningful relationship between the dimensions of resilience and urban sustainability. This study aimed to identify and classify key indicators of resilience in China, as well as to examine and compare these indicators with those of sustainable development. To achieve the desired results, the fuzzy Delphi method was used to identify the indicators, and in the next step, the fuzzy AHP multicriteria decision-making method was used to rank the most important indicators. The findings indicated that out of the five dimensions, the economic indicator, with a weight of 0.41, was the most significant indicator. Additionally, out of the 23 indications, three indicators from this group were deemed to be the most crucial. The second most important indicator was the social and cultural indicator with a weight of 0.194, and the management and institutional indicator took third place with a weight of 0.194. The structural–physical and environmental dimensions were ranked fourth and fifth, respectively.

**Funding:** This research received no external funding.

**Informed Consent Statement:** Not applicable.

**Data Availability Statement:** Not available.

**Conflicts of Interest:** There are no conflict of interest.

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
