# Peer review of "Identifying and Ranking the Dimensions of Urban Resilience and Its Effect on Sustainable Urban Development in Tongdejie, China"

_sustainability, doi:10.3390/su15065606_

Round 1

Reviewer 1 Report

The paper is well organized and sections are sufficiently developed. 

1- It needs, however, extensive editing of English language. 

2- Also, the significance of the topic needs to be clearly stated in the introduction.  One would expect that there is a relationship between the dimensions of resilience and urban sustainability without reading this paper.  In other words, the novelty of the paper is questionable. We need to know why this relationship is significant for urban planning or urban management.  

3- We are informed that the criteria and sub-criteria are identified through a "comprehensive review of the theoretical foundations".  However, I could not find any clear reference to or review of literature which point to these indicators.  May be a table can help which shows where these indicators come from.

4- It is not clear who the "decision-making experts" are:  Their affiliations, and why 12 of them?  Why not more or less?

Author Response

Reviewer 1

The paper is well organized and sections are sufficiently developed. 

  • It needs, however, extensive editing of English language.

Response: Thank you for this valuable comment. We double-checked the English throughout the manuscript with a native speaker. Further necessary corrections can be done during the language editors of the journal.

  • Also, the significance of the topic needs to be clearly stated in the introduction.  One would expect that there is a relationship between the dimensions of resilience and urban sustainability without reading this paper.  In other words, the novelty of the paper is questionable. We need to know why this relationship is significant for urban planning or urban management.

Response: Thank you for this valuable comment. The suggestions are quite relevant. Added.

Both theoretically and empirically, there is an interesting relationship between resilience and urban sustainability. The resilience of an urban area contributes to its sustainability and leads to sustainable solutions in the process of sustainable development. To achieve the desired future, it is essential to pay attention to the benefits of urban resilience in order to strengthen sustainable urban systems [22]; Furthermore, urban sustainability is closely related to the paradigm of sustainable urban development that we introduced in our joint report. Sustainability and resilience are viewed by some as interchangeable concepts, while resilience is viewed by others as one of the goals of sustainability and as its underlying condition  

3- We are informed that the criteria and sub-criteria are identified through a "comprehensive review of the theoretical foundations".  However, I could not find any clear reference to or review of literature which point to these indicators.  May be a table can help which shows where these indicators come from.

Response: Thank you for this valuable comment. Amended

The corresponding indicators and components used in the research are displayed based on the opinions of elites and previous research literature Based on the researches of Norris et al[33]., Patel and Nosal 2016 [34], cimellaro 2016[35], Romero-Lankao et al. 2016[36], Zhang and Li 2018 [7], Ray and Shaw. 2018, Ajibid 2017[37], Borsekova [38].

4- It is not clear who the "decision-making experts" are:  Their affiliations, and why 12 of them?  Why not more or less?

Response: Thank you for this valuable comment. Amended

Fuzzy EHP method is an expert-centered method, which means that the questionnaires should be done by the elites of the target group, and the usual number for this type of questionnaire, which is a pairwise comparison, is between 8- 15 people. The selection of 12 experts for us was based on work experience of more than ten years, related fields of study, i.e. architecture, civil engineering, construction management and urban planning. We were able to access to 12 experts.

Researchers in this study primarily interviewed city planning and municipal employees as well as construction engineers The process begins with a list of experts. After that, 12 experts were purposefully selected based on their education and work experience in order to determine the criteria based on their degree of work experience. Generally, the steps related to the research are given in Figure 1.

Reviewer 2 Report

The author conducted a case study in Tongdejie, China, using Fuzzy AHP to identify and rank the importance of factors that affect urban resilience and sustainable urban development. Overall, the manuscript is very difficult to follow. There are too many unexplained acronyms and grammatical errors. Sending the manuscript to English editors is a must. A few more specific comments are as follows:

(1)   The manuscript should be condensed. There are too much irrelevant information and background introduction. For example, on page 1, line 43, the explanation of the Latin root of resilience is not helpful at all to augment the scientific argument of the manuscript. Another example is Table 3 and Figure 3. The information provided in Figure 3 is shown in Table 3 already.

(2)   There is no connection between the reviewed literature and the proposed research. What does the author learn from the literature? Why is the research necessary in the field of urban sustainability and resilience? More relevant scientific articles should be included to show the innovation and scientific contribution of the manuscript.

(3)   How the author selected the 28 sub-criterion is unclear.

(4)   The Discussion section should focus on discussing the results instead of providing another “history” and literature review.

(5)   The sections after the conclusion section are incomplete.

Author Response

Reviewer 2

The author conducted a case study in Tongdejie, China, using Fuzzy AHP to identify and rank the importance of factors that affect urban resilience and sustainable urban development. Overall, the manuscript is very difficult to follow. There are too many unexplained acronyms and grammatical errors. Sending the manuscript to English editors is a must. A few more specific comments are as follows:

(1) The manuscript should be condensed. There are too much irrelevant information and background introduction. For example, on page 1, line 43, the explanation of the Latin root of resilience is not helpful at all to augment the scientific argument of the manuscript. Another example is Table 3 and Figure 3. The information provided in Figure 3 is shown in Table 3 already.

Response: Thank you for this valuable comment.  Upon your request, dear reviewer, the mentioned sentence was deleted, and we also checked for possible similar cases throughout the entire paper. However, I must point out that there is a difference between Figure 3 and Table 3. Table 3 shows the fuzzy and non-fuzzy scores of the sub-criteria, as well as the sub-criteria that have been removed. As can be seen in Figure 3, we have categorized and schematically displayed the final criteria after the previous step(Table3), which was the calculation of the elite points.

(2)   There is no connection between the reviewed literature and the proposed research. What does the author learn from the literature? Why is the research necessary in the field of urban sustainability and resilience? More relevant scientific articles should be included to show the innovation and scientific contribution of the manuscript.

Response: Thank you for this valuable comment. Two paragraphs are added to current literature review. In the research literature, we reviewed two types of research in general. The first one was the researches that dealt with the selection of resilience and sustainability dimensions, and in the second part, we dealt with the relationship between these two issues and their importance. We also mentioned the work and research location so that the reader's mind will be clear in terms of the importance and relevance of the topic.

The concept of resilience has been integrated into sustainability through several quantitative methods. Through the use of probabilistic risk analysis, Walker et al. (2010)[11] incorporated re-silience into sustainability quantification by defining sustainability as the ability to achieve a non-decreasing level of welfare. A metric-based framework is proposed by Jarzebski et al. (2016) [12]to measure economic, environmental (natural capital), and social (socio-cultural capital) sus-tainability. Resilience indicators include trust in local government, traditional farming practices, agroforestry practices, and forest cover percent. According to Milman and Short (2008)[13], water system sustainability can be built using indicators such as water supply over the next 50 years, quality of service (e.g., chlorinated pipes or public wells), and financial risk to water providers.

There are some studies that include other framework components in addition to resilience as a component of sustainability. According to Saunders and Becker (2015)[14], risk management contributes to sustainability through resilience. Research on resilience includes risk in a variety of ways [15],[16],[17]. By applying a risk management framework to case studies of earth-quake-prone communities in New Zealand, Saunders and Becker (2015)[14] conclude that lower-ing risk leads to greater resilience and sustainability. Similarly, Seager (2008)[18] presents resil-ience as one perspective among the four aspects of sustainability, which also include security, reliability, and renewal. Throughout this spectrum, sustainability moves from a state of security or stability (a state in which the status quo remains unchanged) to one in which rapid change and an all-encompassing reorganization are enabled (Seager, 2008)[18].

(3)   How the author selected the 28 sub-criterion is unclear.

Response: Thank you for this valuable comment. This concern was also risen by other reviewers and we appreciate that.

First, in light of a literature review and ELIT's opinion, the dimensions of urban resilience appear to be as follows: then a questionnaire was given to 12 experts to rate each criterion based on the range of 1 to 5 in Table 1, then based on the fuzzy Delphi method, confirmation and screen-ing The indicators were made and the results are given in Table 3. The results show the rejection of 5 indicators that have obtained a non-fuzzy score of less than 0.7 and 23 indicators have been approved.

 The corresponding indicators and components used in the research are displayed based on the opinions of elites and previous research literature Based on the researches of Norris et al[33]., Patel and Nosal 2016 [34], cimellaro 2016[35], Romero-Lankao et al. 2016[36], Zhang and Li 2018 [7], Ray and Shaw. 2018, Ajibid 2017[37], Borsekova [38].

(4)   The Discussion section should focus on discussing the results instead of providing another “history” and literature review.

We had three objectives in our research, the first being to identify the criteria in five dimensions using both the literature and the opinions of the elites, and then to rank them. The final objective of this study was to analyze the relationship between each of the dimensions and sustainable development indicators in relation to each other. Specifically, our objective was to evaluate and compare these criteria with those previously published by other researchers regarding sustainability. Consequently, at this stage, we had a general definition of the indicators (criteria and sub criteria) that we had selected, and we analyzed these indicators and their significance from the standpoint of sustainability.

(5)   The sections after the conclusion section are incomplete.

Reviewer 3 Report

Thanks for the review invitation. I only have one minor comment. It might be better to underscore the research innovations in the introduction. In this current version, the innovation is not clear.

Author Response

Reviewer 3

Thanks for the review invitation. I only have one minor comment. It might be better to underscore the research innovations in the introduction. In this current version, the innovation is not clear.

Response: Thank you for this valuable comment. The suggestions are quite relevant. Amended.

The purposes of this study are as follows, first by identifying the criteria within five dimensions based on both literature and the opinions of elites, and then by ranking them. Afterwards, this study aims to analyze the relationship between each of the dimensions and sustainable development indicators. As a final objective, we intend to compare our criteria with those previously published by other researchers regarding sustainability, and to analyze the importance of these indicators from a sustainability perspective. These results provide policymakers and city officials with a clue for direction and a greater awareness of obstacles to urban resilience and sustainability

Reviewer 4 Report

1- Please add a clear definition for resilience in your application area. The term has different meanings in various fields. 

2- I suggest separating the literature review from the introduction. The current introduction section is hard to follow. In addition, you can highlight the gaps you covered in your study. 

3- Please revise Figure 1. General stages of research. The text in one of the boxes is not complete. 

4- Please change the underlined sentence in 2.2. Fuzzy Delphi method section to normal text. 

5- Please add the limits for summation in equations 2-4. You may change the notation for the crisp value, as well. 

6-Some equations should be revised (e.g., equations 7 and 8). Please revise the formulations and make them consistent throughout the paper.  

7- You may add a reference to the criteria you adopted from the literature in Table 3. Fuzzy score sub-criterion Non-fuzzy status score.

8- You also need to add a brief description regarding the criteria such as city form. 

9- The contribution of the paper is unclear to me. How is the proposed method different than the current ones in the literature? Please clarify this part. 

Author Response

Reviewer 4

  • Please add a clear definition for resilience in your application area. The term has different meanings in various fields. 

Response: Thank you for this valuable comment. Amended.

It is necessary to have knowledge of how the theory of resilience was developed in order to comprehend the concept of urban resilience, even though it has long been applied in psychology and engineering. When it comes to global literature about environmental changes, resilience is typically referred to by researches that date back to Holling's name. The concept of resilience has been defined by Holling as the capability of systems to maintain their basic functions in the face of disruptions.[9] He distinguishes between static (passive) engineering resilience (referring to the ability of a system to return to its previous state) and dynamic ecological resilience (referring to the ability of key functions to be maintained during disturbances) by describing ecosystems as multiple stable states[10]. Our research employs this meaning and examines resilience in China's Tongdejie region in terms of five economic, social, and cultural dimensions, environmental factors, physical structure, and management issues.

  • I suggest separating the literature review from the introduction. The current introduction section is hard to follow. In addition, you can highlight the gaps you covered in your study. 

Response: Thank you for this valuable comment. The suggestions are quite relevant. Added.

  • Please revise Figure 1. General stages of research. The text in one of the boxes is not complete. 

Response: Thank you for this valuable comment. The mentioned figure was modified.

4- Please change the underlined sentence in 2.2. Fuzzy Delphi method section to normal text. 

Response: Thank you for this valuable comment. The text was corrected.

5- Please add the limits for summation in equations 2-4. You may change the notation for the crisp value, as well. 

Response: Thank you for this valuable comment. Amended.

6-Some equations should be revised (e.g., equations 7 and 8). Please revise the formulations and make them consistent throughout the paper.  

Response: Thank you for this valuable comment. The requested items are addressed

7- You may add a reference to the criteria you adopted from the literature in Table 3. Fuzzy score sub-criterion Non-fuzzy status score.

Response: Thank you for this valuable comment. Amended.

The corresponding indicators and components used in the research are displayed based on the opinions of elites and previous research literature Based on the researches of Norris et al[33]., Patel and Nosal 2016 [34], cimellaro 2016[35], Romero-Lankao et al. 2016[36], Zhang and Li 2018 [7], Ray and Shaw. 2018, Ajibid 2017[37], Borsekova [38].

8- You also need to add a brief description regarding the criteria such as city form.

Response: Thank you for this valuable comment. We had a general definition of the indicators (criteria and sub criteria) that we had selected, and we analyzed these indicators and their significance from the standpoint of sustainability and resiliency in the discussion section.

The physical-structural category includes uses, texture and body of the city, city form, buildings and historical buildings and neighborhood cohesion[51]. The resilience of the place does not only refer to possible operations such as quick response to critical situations or events such as earthquakes, floods and other disasters in vulnerable areas, but also includes adaptive strategies and long-term reduction of crisis effects in facing environmental and social challenges[52]

9- The contribution of the paper is unclear to me. How is the proposed method different than the current ones in the literature? Please clarify this part. 

Response: Thank you for this valuable comment. Regarding the research done, it can be said that the majority of the research has been conducted to investigate resilience using a series of specific criteria, however, in the current study, the relevant criteria were first identified from research literature and then screened using fuzzy Delphi. The indicators were screened based on elit's opinion, which resulted in the removal of five indicators. Finally, these indicators were ranked according to the opinion of the elite. Furthermore, we examined these indicators and their relationship and importance with sustainability indicators at the end of the article, which makes it an innovative contribution.

Round 2

Reviewer 2 Report

(1)   The author's response to comment (1): “As can be seen in Figure 3, we have categorized and schematically displayed the final criteria after the previous step(Table3), which was the calculation of the elite points.” Table 3 already provides sufficient information by displaying the status of "confirm" and "reject". The author's response does not justify the need for Figure 3, which seems redundant. Also, the font size of Figure 3 is also too small to be easily readable.

Additionally, the manuscript still contains excessive background information and does not adequately address the reviewer's concerns about the clarity of the scientific contribution.

(2)   The author's response to comment (2) by adding more literature review does not address the reviewer's concerns about the connection between the literature and the study and how it contributes to the field of urban sustainability and resilience. The inclusion of additional literature summaries without clear connections to the study does not improve the readers' understanding of the novelty and scientific significance of the manuscript.

(3)   The author's response to comment (4) explains the reasoning for keeping all information in the discussion section, but it does not address the reviewer's concern. For example, the reviewer believes that the first three paragraphs of the discussion section in the revised manuscript would be more appropriately placed in the introduction section as they are not directly related to the discussion.

(4)   The author contribution, funding, data availability, acknowledgments, and conflicts of interest sections are ALL incomplete.

(5)   The manuscript requires extensive English editing. The author's response that the manuscript was double-checked by a native speaker and further corrections will be made during the journal's language editing process does not alleviate the reviewer's concern about the difficulty in understanding the manuscript due to poor English. The current state of English makes it difficult for the reviewer to fully comprehend the manuscript.

(6)   The reference style in the manuscript is not consistent. The author should choose one reference format and ensure consistency throughout the manuscript.

(7)   The use of "we" in the author's responses and the revision raises questions about the number of contributors to the manuscript. If there are multiple contributors, they should either be listed as authors or acknowledged in some way.

Author Response

Reviewer 2

  • The author's response to comment (1): “As can be seen in Figure 3, we have categorized and schematically displayed the final criteria after the previous step(Table3), which was the calculation of the elite points.” Table 3 already provides sufficient information by displaying the status of "confirm" and "reject". The author's response does not justify the need for Figure 3, which seems redundant. Also, the font size of Figure 3 is also too small to be easily readable. Additionally, the manuscript still contains excessive background information and does not adequately address the reviewer's concerns about the clarity of the scientific contribution.

Response: Thank you for this valuable comment. Amended. Regarding figure 3 and table 3, according to your request, dear referee, I have removed figure 3. Also, regarding the second part of your comment, which is related to the unnecessary parts in relation to scientific participation, I provided detailed explanations in the second comment along with the literature review.

(2)   The author's response to comment (2) by adding more literature review does not address the reviewer's concerns about the connection between the literature and the study and how it contributes to the field of urban sustainability and resilience. The inclusion of additional literature summaries without clear connections to the study does not improve the readers' understanding of the novelty and scientific significance of the manuscript.

Response: Thank you for this valuable comment. I checked the literature again. With respect, I would like to explain a few points regarding the research literature that I have written. First of all, in the first part, i.e., the first two paragraphs, I mentioned the researches done on the relationship between sustainability and resilience which these articles in recent years have focused on the integration and connection of these two concepts. In the next paragraphs, I mentioned the researches that were done to evaluate resilience according to different indicators and dimensions in different countries. Finally, according to your request, I have added a few more studies that have only focused on the selection and identification of resilience dimensions. This means that I have mentioned the studies that are related to my research, the studies that are related to resilience and sustainability related to urban issues and urban planning, as well as the studies that are related to my research. They have identified the dimensions of resilience, indicators, and how to evaluate resilience. I mentioned that these were the 3 main goals of my research. I also feel it necessary to state that the last four paragraphs of the literature have mentioned the goals of research and innovation and the existing research gap.

  • The author's response to comment (4) explains the reasoning for keeping all information in the discussion section, but it does not address the reviewer's concern. For example, the reviewer believes that the first three paragraphs of the discussion section in the revised manuscript would be more appropriately placed in the introduction section as they are not directly related to the discussion.

Response: Thank you for this valuable comment. The suggestions are quite relevant. Amended.

(4)   The author contribution, funding, data availability, acknowledgments, and conflicts of interest sections are ALL incomplete.

Response: Thank you for this valuable comment. The requested items are addressed.

(5)   The manuscript requires extensive English editing. The author's response that the manuscript was double-checked by a native speaker and further corrections will be made during the journal's language editing process does not alleviate the reviewer's concern about the difficulty in understanding the manuscript due to poor English. The current state of English makes it difficult for the reviewer to fully comprehend the manuscript.

Response: Thanks for the valuable comment although I checked the English throughout the manuscript with a native speaker. I reviewed the text again and corrected the errors. Thank you for your kind attention. More necessary corrections can be made during the language editors of the Journal.

(6)   The reference style in the manuscript is not consistent. The author should choose one reference format and ensure consistency throughout the manuscript.

Response: Thank you for this valuable comment. The style of references was reviewed and modified.

(7)   The use of "we" in the author's responses and the revision raises questions about the number of contributors to the manuscript. If there are multiple contributors, they should either be listed as authors or acknowledged in some way.

Response: Thank you for this valuable comment. I am the only author of my article. Thank you for your attention. The text was thoroughly checked and this issue was resolved.

Reviewer 4 Report

I would like to thank the authors for revising the paper based on my comments. I feel the paper is much improved and can be published. 

Author Response

Reviewer 4

I would like to thank the authors for revising the paper based on my comments. I feel the paper is much improved and can be published. 

Response: Thank you for your kind attention, dear Reviewer.
